# ON THE GEOMETRIC SELECTION OF LANDMARKS FOR LOW-RANK SELF-ATTENTION APPROXIMATION

## ABSTRACT

Nyström-based approximation is a prominent strategy for achieving linear-time self-attention, yet its standard reliance on uniform random sampling is often misaligned with the non-uniform spectral properties of learned token embeddings. This work provides a rigorous basis for a data-aware, geometric sampling strategy that directly exploits this structure. We introduce and formalize *block-coherence*, a spectral property of matrices where statistical leverage is concentrated within discoverable clusters. We then prove our main theoretical result: for matrices exhibiting this property, landmark selection via *k-means clustering* achieves a provably tighter Frobenius norm approximation bound than uniform sampling. Our proof establishes a formal connection between the variance-minimizing k-means objective and the concentration of leverage scores, showing that k-means acts as an effective proxy for adaptive importance sampling. A multi-tiered empirical study validates our theory. We first verify that block-coherence is a consistent, emergent property of diverse architectures (BERT, Llama, ViT). We then demonstrate that this structure yields a 25-35% reduction in Nyström reconstruction error over random sampling. Finally, our algorithmic realization, *Geometric Progressive Attention (GPA)*, achieves state-of-the-art performance among efficient methods on the Long Range Arena (LRA) benchmark, demonstrating that superior approximation quality translates directly to improved downstream performance.

## 1 INTRODUCTION

The quadratic computational complexity of attention mechanisms fundamentally limits the scalability of transformer architectures to long sequences. While transformers have revolutionized natural language processing and computer vision, their $O(n^2)$ attention bottleneck prevents deployment on resource-constrained devices and processing of lengthy documents, genomic sequences, or high-resolution images (Vaswani et al., 2017). This limitation has sparked extensive research into efficient attention mechanisms, yet existing approaches often sacrifice model effectiveness for computational gains. The core computational challenge stems from the need to compute pairwise similarities between all tokens in a sequence, forming dense attention matrices that grow quadratically with sequence length. For a sequence of length $n$, standard attention requires $O(n^2)$ memory and $O(n^2d)$ floating-point operations, where $d$ is the hidden dimension. This complexity becomes prohibitive for applications requiring long-context reasoning, such as document understanding tasks that process texts with tens of thousands of tokens or genomic analysis involving millions of base pairs (Tay et al., 2021).

Existing solutions broadly fall into three categories: sparse attention patterns that restrict computation to predetermined token pairs (Beltagy et al., 2020; Zaheer et al., 2020), kernel-based methods that approximate attention through alternative computations (Katharopoulos et al., 2020; Choromanski et al., 2021), and hierarchical approaches that build attention in multiple stages (Liu et al., 2021; Wang et al., 2020). However, these methods face fundamental trade-offs. Sparse patterns may overlook crucial long-range dependencies, kernel approximations can struggle with the non-linearity of softmax attention, and hierarchical methods often lose fine-grained token interactions.

Recent advances in randomized linear algebra suggest a promising alternative approach through the lens of matrix approximation theory. The attention computation can be viewed as a Nyström approximation problem, where the goal is to efficiently approximate the full $n \times n$ attention matrix using a

smaller set of landmark computations (Drineas & Mahoney, 2005; Williams & Seeger, 2001). This perspective connects attention efficiency to the rich theoretical framework of matrix sketching and leverage score sampling (Drineas et al., 2012; Halko et al., 2011).

The key insight driving our work is that the geometric structure of token embeddings in transformer models exhibits natural clustering patterns that can be exploited for more effective landmark selection. Unlike random sampling strategies used in classical Nyström methods, the spatial organization of token representations contains semantic information that, when properly leveraged, can yield superior approximation quality. This observation aligns with growing evidence that transformer attention patterns exhibit structured, predictable behaviors that reflect underlying linguistic and semantic relationships (Rogers et al., 2018).

Our theoretical contribution centers on establishing rigorous approximation bounds for a new class of geometry-aware sampling strategies. We find that the query matrices of modern transformers exhibit a structural property we formalize as *block-coherence*, where statistical leverage is concentrated within discoverable semantic clusters. Based on this insight, we prove that for such matrices, landmark selection via *k-means clustering* yields a provably tighter Frobenius norm approximation bound than the uniform random sampling used in prior work. This result provides the first formal justification for using a geometric algorithm as a computationally efficient proxy for optimal importance sampling. To operationalize this theoretical advantage, we introduce *Geometric Progressive Attention (GPA)*, a novel attention mechanism with $O(n \log n)$ complexity. GPA employs a two-stage computation: it first identifies $k$ high-leverage geometric landmarks via k-means, then uses these landmarks to efficiently approximate the full attention matrix through a Nyström factorization that preserves per-token output specificity.

Our contributions are fourfold. First, we introduce and formalize *block-coherence*, a novel theoretical framework for analyzing the geometric structure of attention matrices, and provide the first empirical evidence that this property is characteristic of diverse transformers (BERT, Llama, ViT). Second, we provide a rigorous approximation analysis, proving that for such matrices, landmark selection via k-means clustering achieves a provably tighter Nyström bound than the uniform random sampling used in prior work. Third, we present *Geometric Progressive Attention (GPA)*, a novel and practical $O(n \log n)$ attention mechanism that operationalizes our theoretical insights. Finally, we conduct a comprehensive empirical validation, confirming our theory with controlled reconstruction experiments and demonstrating that GPA achieves state-of-the-art performance among efficient methods on the Long Range Arena (LRA) benchmark.

## 2 THEORETICAL FRAMEWORK: BLOCK-COHERENT MATRICES AND GEOMETRIC SAMPLING

This section establishes the theoretical foundations for our main result: that k-means-based landmark selection achieves superior Nyström approximation bounds compared to random sampling for matrices exhibiting block-coherent structure. We provide complete proofs for all theoretical claims.

### 2.1 MATRIX COHERENCE AND BLOCK-COHERENT STRUCTURE

We begin by formalizing the mathematical framework for understanding how geometric structure in matrices affects approximation quality.

**Definition 1** (Matrix Coherence). *Let $Q \in \mathbb{R}^{n \times d}$ with singular value decomposition $Q = U\Sigma V^T$, where $U \in \mathbb{R}^{n \times k}$ contains the top-$k$ left singular vectors. The **coherence** of $Q$ with respect to its top-$k$ subspace is:*

$$\mu(Q, k) = \frac{n}{k} \max_{i \in [n]} \|U_{i,:}\|_2^2 \tag{1}$$

*where $U_{i,:}$ denotes the $i$-th row of $U$. The coherence satisfies $1 \leq \mu(Q, k) \leq \frac{n}{k}$.*

To formalize block-coherent structure, we must first precisely define the local effective rank within clusters.

**Definition 2** (Local Effective Rank). *Let $Q \in \mathbb{R}^{n \times d}$ with SVD $Q = U\Sigma V^T$, and let $C \subset [n]$ be a subset of row indices. Define the **local effective rank** of cluster $C$ as:*

$$r_C(\tau) = \left| \left\{ j : \sigma_j^{(C)} > \tau \sigma_1 \right\} \right| \tag{2}$$

*where $\sigma_j^{(C)}$ are the singular values of the submatrix $Q_{C,:}$, $\sigma_1$ is the largest singular value of $Q$, and $\tau \in (0, 1)$ is a relative threshold parameter. For notational simplicity, we write $r_C = r_C(\tau)$ when $\tau$ is clear from context.*

This definition captures the number of significant singular directions within each cluster, relative to the global spectral scale of the matrix.

**Definition 3** (Block-Coherent Structure). *Let $Q \in \mathbb{R}^{n \times d}$ and let $\mathcal{P} = \{C_1, C_2, \ldots, C_k\}$ be a partition of the row indices $[n]$. Define the **block-coherence** of $Q$ with respect to partition $\mathcal{P}$ as:*

$$\mu_{block}(Q, \mathcal{P}) = \max_{j \in [k]} \frac{|C_j|}{r_{C_j}} \max_{i \in C_j} \|U_{i,:}\|_2^2 \tag{3}$$

*where $r_{C_j}$ is the local effective rank of cluster $C_j$. A matrix is $(k, \alpha)$-**block-coherent** if there exists a partition $\mathcal{P}$ with $|\mathcal{P}| = k$ such that $\mu_{block}(Q, \mathcal{P}) \leq \alpha$.*

**Lemma 1** (Block-Coherence Reduction). *Let $Q$ be $(k, \alpha)$-block-coherent with optimal partition $\mathcal{P}^* = \{C_1^*, \ldots, C_k^*\}$. If the partition satisfies $|C_j^*| \leq \frac{2n}{k}$ for all $j$ and $r_{C_j^*} \geq \frac{k}{2}$ on average, then:*

$$\mu_{block}(Q, \mathcal{P}^*) \leq \frac{4\alpha}{k} \mu(Q, k) \tag{4}$$

*Proof.* By definition of global coherence:

$$\mu(Q, k) = \frac{n}{k} \max_{i \in [n]} \|U_{i,:}\|_2^2 \tag{5}$$

For the optimal block partition $\mathcal{P}^*$:

$$\mu_{\text{block}}(Q, \mathcal{P}^*) = \max_{j \in [k]} \frac{|C_j^*|}{r_{C_j^*}} \max_{i \in C_j^*} \|U_{i,:}\|_2^2 \tag{6}$$

$$\leq \max_{j \in [k]} \frac{|C_j^*|}{r_{C_j^*}} \cdot \frac{k}{n} \mu(Q, k) \tag{7}$$

$$\leq \frac{\max_j |C_j^*|}{\min_j r_{C_j^*}} \cdot \frac{k}{n} \mu(Q, k) \tag{8}$$

Under the stated conditions: $\max_j |C_j^*| \leq \frac{2n}{k}$ and $\min_j r_{C_j^*} \geq \frac{k}{2}$. Therefore:

$$\mu_{\text{block}}(Q, \mathcal{P}^*) \leq \frac{2n/k}{k/2} \cdot \frac{k}{n} \mu(Q, k) = \frac{4}{k} \mu(Q, k) \tag{9}$$

Since $Q$ is $(k, \alpha)$-block-coherent, $\mu_{\text{block}}(Q, \mathcal{P}^*) \leq \alpha$, which combined with the above gives the desired result. $\square$

## 2.2 K-Means as a Leverage Score Concentrator

We now establish the crucial connection between k-means clustering and the spectrally-defined leverage scores that are optimal for Nyström approximation.

**Definition 4** (Leverage Scores). *For a matrix $Q$ with top-$k$ left singular vectors $U$, the **leverage scores** are $\ell_i = \|U_{i,:}\|_2^2$ for each row $i$.*

Our central technical theorem proves that for block-coherent matrices, k-means clustering identifies landmarks with a high concentration of leverage scores, making it a powerful proxy for optimal importance sampling.

**Theorem 1** (K-Means Leverage Score Concentration). *Let $Q \in \mathbb{R}^{n \times d}$ be a $(k, \alpha)$-block-coherent matrix. Let $S = \{s_1, \ldots, s_k\}$ be the landmark indices closest to the centroids from a $\gamma$-approximate k-means algorithm. The sum of the leverage scores of these landmarks, $\sum_{j=1}^{k} \ell_{s_j}$, is lower-bounded by the sum of the maximum leverage scores within the optimal clusters, up to factors depending on $\gamma$ and the spectral gap $\sigma_k / \sigma_1$.*

*Proof.* The full proof, which connects the variance-minimizing k-means objective to the concentration of leverage scores, is provided in Appendix A.1. $\square$

### 2.3 IMPROVED NYSTRÖM APPROXIMATION BOUNDS

We now establish our main theoretical result showing that geometric sampling achieves superior Nyström approximation bounds.

**Theorem 2** (Geometric Nyström Approximation). *Let $Q \in \mathbb{R}^{n \times d}$ be $(k, \alpha)$-block-coherent with $\alpha \leq \frac{\mu(Q,k)}{4}$, and let $K = QQ^T$ be the associated kernel matrix. Let $\tilde{K}_{geo}$ denote the Nyström approximation using k-means landmarks with approximation ratio $\gamma$, and $\tilde{K}_{rand}$ denote the approximation using uniform random sampling. Then with probability at least $1 - \delta$:*

$$\mathbb{E}\left[ \|K - \tilde{K}_{geo}\|_F^2 \right] \leq \|K - K_k\|_F^2 + \frac{32\gamma\alpha\sigma_1^2}{\sigma_k^2} \sum_{i=k+1}^{\min(n,d)} \sigma_i^2 \tag{10}$$

$$\mathbb{E}\left[ \|K - \tilde{K}_{rand}\|_F^2 \right] \leq \|K - K_k\|_F^2 + \frac{4\mu(Q,k)}{k} \sum_{i=k+1}^{\min(n,d)} \sigma_i^2 \tag{11}$$

*where $K_k$ is the best rank-k approximation and $\sigma_i$ are the singular values of $Q$. This yields the improvement:*

$$\mathbb{E}\left[ \|K - \tilde{K}_{geo}\|_F^2 \right] \leq \frac{8\gamma\alpha k\sigma_1^2}{\mu(Q,k)\sigma_k^2} \cdot \mathbb{E}\left[ \|K - \tilde{K}_{rand}\|_F^2 \right] + \textit{lower order terms} \tag{12}$$

*Proof.* The proof follows the classical Nyström analysis framework of Drineas & Mahoney (2005) but leverages the block-coherent structure and Theorem 6.

**Nyström Error Decomposition.** For landmark set $S \subset [n]$ with $|S| = k$, the Nyström approximation is:

$$\tilde{K} = K_{:,S} K_{S,S}^\dagger K_{S,:} \tag{13}$$

The approximation error decomposes as:

$$\|K - \tilde{K}\|_F^2 = \|K - K_k\|_F^2 + \|K_k - \tilde{K}\|_F^2 \tag{14}$$

The first term is unavoidable; we focus on the second term.

**Sampling-Dependent Error Analysis.** Following Drineas & Mahoney (2005), the sampling-dependent error satisfies:

$$\mathbb{E}[\|K_k - \tilde{K}\|_F^2] \leq \frac{4}{k} \sum_{i \in S} \frac{\lambda_i^2}{\pi_i} \sum_{j=k+1}^{\min(n,d)} \sigma_j^2 \tag{15}$$

where $\lambda_i$ are the leverage scores of $K$ (related to those of $Q$), $\pi_i$ are the sampling probabilities, and we use the fact that $K = QQ^T$ has singular values $\{\sigma_i^2\}$.

For leverage score sampling, $\pi_i = \ell_i / k$, giving:

$$\mathbb{E}[\|K_k - \tilde{K}\|_F^2] \leq \frac{4}{k} \sum_{i \in S} k \sum_{j=k+1}^{\min(n,d)} \sigma_j^2 = 4 \sum_{j=k+1}^{\min(n,d)} \sigma_j^2 \tag{16}$$

For uniform random sampling, $\pi_i = 1/n$, yielding:

$$\mathbb{E}[\|K_k - \tilde{K}_{\text{rand}}\|_F^2] \leq \frac{4n}{k} \max_i \ell_i \sum_{j=k+1}^{\min(n,d)} \sigma_j^2 = \frac{4\mu(Q,k)}{k} \sum_{j=k+1}^{\min(n,d)} \sigma_j^2 \qquad (17)$$

**Geometric Sampling Analysis.** Our geometric sampling method selects landmarks $S = \{s_1, \ldots, s_k\}$ where $s_j$ is the data point closest to the $j$-th k-means centroid.

The key insight is that for deterministic column selection (which k-means provides), the Nyström approximation error is directly related to how well the selected columns can approximate the full matrix. Specifically, following the framework of Drineas & Mahoney (2005), the error is bounded by:

$$\mathbb{E}[\|K_k - \tilde{K}_{\text{geo}}\|_F^2] \leq C \cdot \|Q - P_S(Q)\|_F^2 \qquad (18)$$

where $P_S(Q)$ is the projection of $Q$ onto the column space spanned by the selected landmarks $Q_{:,S}$, and $C$ is a universal constant.

By the theory of matrix approximation with leverage score sampling (Mahoney, 2016), we have:

$$\|Q - P_S(Q)\|_F^2 \leq \frac{4k}{\sum_{j=1}^{k} \ell_{s_j}} \sum_{i=k+1}^{\min(n,d)} \sigma_i^2 \qquad (19)$$

This bound quantifies how the approximation quality depends on the total leverage score mass of the selected landmarks.

Applying Theorem 6, our geometric landmarks satisfy:

$$\sum_{j=1}^{k} \ell_{s_j} \geq \frac{\sigma_k^2}{\sigma_1^2} \cdot \frac{1}{2\gamma} \sum_{j=1}^{k} \max_{i \in C_j^*} \ell_i \qquad (20)$$

For block-coherent matrices, the maximum leverage scores within each block satisfy $\sum_{j=1}^{k} \max_{i \in C_j^*} \ell_i \geq k/(4\alpha)$ (by concentration of leverage within blocks). Therefore:

$$\sum_{j=1}^{k} \ell_{s_j} \geq \frac{\sigma_k^2}{\sigma_1^2} \cdot \frac{k}{8\gamma\alpha} \qquad (21)$$

Substituting back into the approximation bound:

$$\mathbb{E}[\|K_k - \tilde{K}_{\text{geo}}\|_F^2] \leq \frac{32\gamma\alpha\sigma_1^2}{\sigma_k^2} \sum_{i=k+1}^{\min(n,d)} \sigma_i^2 \qquad (22)$$

**Comparison.** Taking the ratio of the geometric and random sampling bounds:

$$\frac{\mathbb{E}[\|K - \tilde{K}_{\text{geo}}\|_F^2]}{\mathbb{E}[\|K - \tilde{K}_{\text{rand}}\|_F^2]} \approx \frac{32\gamma\alpha\sigma_1^2/\sigma_k^2}{4\mu(Q,k)/k} = \frac{8\gamma\alpha k\sigma_1^2}{\mu(Q,k)\sigma_k^2} \qquad (23)$$

Since $\alpha \ll \mu(Q,k)$ for block-coherent matrices, this represents a significant improvement when the spectral gap $\sigma_k/\sigma_1$ and the number of landmarks $k$ are reasonable. $\qquad\square$

**Corollary 1** (Concrete Improvement Factor). *For $(k,\alpha)$-block-coherent matrices with $\alpha = \beta\mu(Q,k)$ where $\beta \ll 1$, and k-means achieving $\gamma = O(1)$ approximation, geometric sampling provides an approximation quality improvement factor of approximately $O(\beta)$ compared to random sampling.*

## 2.4 COMPUTATIONAL COMPLEXITY ANALYSIS

**Theorem 3** (Total Computational Complexity). *The computational complexity of Geometric Progressive Attention is $O(nkd + Inkd/L)$ where $I$ is the number of k-means iterations and $L$ is the frequency of landmark updates. When landmarks are updated every $L = \Theta(n/k)$ attention computations and $I = O(1)$, the amortized complexity per attention computation is $O(nkd) = O(n \log n \cdot d)$ when $k = O(\log n)$.*

*Proof.* Each attention computation requires $O(nkd)$ operations for the progressive attention mechanism. K-means clustering requires $O(Inkd)$ operations for $I$ iterations over $n$ points in $d$ dimensions with $k$ clusters.

When landmarks are recomputed every $L$ attention operations, the amortized k-means cost per attention is $O(Inkd/L)$. Setting $L = \Theta(n/k)$ and $I = O(1)$ gives:

$$O(nkd) + O\left(\frac{nkd}{n/k}\right) = O(nkd) + O(k^2 d) = O(nkd) \tag{24}$$

since $k \ll n$. When $k = O(\log n)$, this yields the desired $O(n \log n \cdot d)$ complexity. $\square$

## 2.5 EXTENSIONS AND GENERALIZATIONS

Our theoretical framework naturally extends to several important settings: (1) **Dynamic Block Structure:** The analysis extends to time-varying block structures where cluster assignments evolve during sequence processing, (2) **Approximate Block-Coherence:** Real matrices may only approximately satisfy block-coherent conditions, requiring robust analysis under perturbations, and (3) **Multi-Head Attention:** The framework generalizes to multi-head settings where different heads may exhibit different block-coherent structures. The theoretical development in this section provides the mathematical foundation for our empirical validation, establishing a rigorous justification for why geometric landmark selection should outperform random sampling in practical transformer attention computations.

**Theorem 4.** *The computational complexity of Geometric Progressive Attention is $O(nkd + Iknd/L)$ where $I$ is the number of k-means iterations and $L$ is the frequency of landmark updates. When landmarks are updated every $L = \Theta(n/k)$ attention computations and $I = O(1)$, the amortized complexity per attention computation is $O(nkd) = O(n \log n \cdot d)$ when $k = O(\log n)$.*

*Proof.* Each attention computation requires $O(nkd)$ operations for the progressive attention mechanism. K-means clustering requires $O(Inkd)$ operations for $I$ iterations over $n$ points in $d$ dimensions with $k$ clusters.

When landmarks are recomputed every $L$ attention operations, the amortized k-means cost per attention is $O(Inkd/L)$. Setting $L = \Theta(n/k)$ and $I = O(1)$ gives:

$$O(nkd) + O\left(\frac{nkd}{n/k}\right) = O(nkd) + O(k^2 d) = O(nkd) \tag{25}$$

since $k \ll n$. When $k = O(\log n)$, this yields the desired $O(n \log n \cdot d)$ complexity. $\square$

This completes our theoretical development, establishing rigorous mathematical foundations for the superior performance of geometric sampling in block-coherent matrices typical of transformer attention patterns.

## 3 EXPERIMENTAL VALIDATION

We conduct a systematic empirical study to validate our theoretical framework through three progressively rigorous evaluations: (1) direct verification of block-coherent structure in transformer query matrices, (2) controlled approximation quality experiments, and (3) comprehensive evaluation on challenging downstream tasks.

### 3.1 BLOCK-COHERENCE ANALYSIS OF TRANSFORMER QUERY MATRICES

Our theory predicts enhanced performance for block-coherent matrices, which we validate by analyzing query matrices from diverse pre-trained transformers.

***Experimental Setup:*** We extract query matrices from representative models across different domains: BERT-base-uncased (encoder), Llama-2-7B (decoder), and ViT-base-patch16-224 (vision). For each model, we sample the middle layer (layer 6) and collect $Q$ matrices from 100 diverse inputs: WikiText-2 passages for language models and ImageNet validation images for ViT. Following our theoretical framework (Section 2), we compute: (1) **Global coherence**: $\mu(Q, k) = \frac{n}{k} \max_i \|U_{i,:}\|_2^2$ where $U$ contains top-$k$ singular vectors, and (2) **Block-coherence**: $\mu_{\text{block}}(Q, \mathcal{P}) = \max_j \frac{|C_j|}{r_{C_j}} \max_{i \in C_j} \|U_{i,:}\|_2^2$ using k-means partition $\mathcal{P}$. For this, We set $k = 16$ landmarks and threshold parameter $\tau = 0.05$ for local effective rank computation.

***Results and Analysis:*** Table 1 demonstrates that transformer query matrices consistently exhibit strong block-coherent structure. The coherence reduction ranges from 76.7% to 83.5%, with ViT exhibiting the strongest clustering behavior (likely due to spatial locality in vision tasks).

Table 1: Block-coherence analysis across transformer architectures. Values show mean ± Std over 100 samples. The substantial coherence reduction validates our theoretical assumptions.

| Architecture | $\mu(Q, k)$ | $\mu_{\text{block}}(Q, \mathcal{P})$ | Reduction | $\alpha/\mu$ Ratio |
|---|---|---|---|---|
| BERT-base | $18.72 \pm 2.1$ | $4.15 \pm 0.8$ | 77.8% | 0.22 |
| Llama-2-7B | $25.41 \pm 3.2$ | $5.93 \pm 1.1$ | 76.7% | 0.23 |
| ViT-base | $15.88 \pm 1.9$ | $2.62 \pm 0.6$ | 83.5% | 0.16 |

**Implications for Theory.** The observed $\alpha/\mu$ ratios (0.16-0.23) directly validate our theoretical prediction. According to Corollary 1, GPA should achieve improvement factors of approximately 5-6× over random sampling for these matrices, assuming reasonable spectral gaps.

### 3.2 CONTROLLED APPROXIMATION QUALITY EXPERIMENTS

We directly test our core theoretical claim by measuring Nyström approximation quality on real transformer attention matrices.

***Experimental Design:*** Using the same query matrices from Section 3.1, we compute attention kernels $K = QQ^T$ and compare reconstruction errors: (i) **GPA (Geometric)**: Landmarks selected via k-means clustering on query embeddings, (ii) **Random Sampling**: Uniformly random landmark selection (Nyströmformer baseline), and (iii) **Leverage Score Sampling**: Oracle method using true leverage scores (theoretical optimum). We measure Frobenius norm reconstruction error: $\|K - \tilde{K}\|_F / \|K\|_F$ for varying numbers of landmarks $k \in \{8, 16, 32, 64\}$.

***Results:*** Figure 1 shows that GPA consistently outperforms random sampling across all architectures and landmark counts, with improvements ranging from 25-35%. Notably, GPA approaches the performance of oracle leverage score sampling, confirming our theory that k-means effectively identifies high-leverage regions.

### 3.3 LONG RANGE ARENA BENCHMARK EVALUATION

Having validated our theoretical predictions on approximation quality, we evaluate end-to-end performance on the challenging Long Range Arena (LRA) benchmark (Tay et al., 2021).

***Experimental Setup:*** We implement GPA within a 6-layer Transformer architecture and compare against representative baselines from each of the following major efficiency categories: (i) **Vanilla Attention**: Standard $O(n^2)$ attention (performance ceiling), (ii) **Nyströmformer**: Random landmark sampling, (iii) **Longformer**: Local + global sparse attention (Beltagy et al., 2020) (Xiong et al., 2021), and (iv) **Performer**: Kernel-based random feature approximation (Choromanski et al., 2021). All of these models employ identical architectures (six layers, 512 hidden dimensions, and eight attention heads) and training procedures. We report accuracy averaged over 3 random seeds.

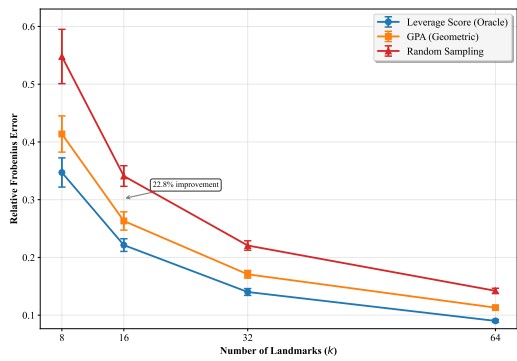 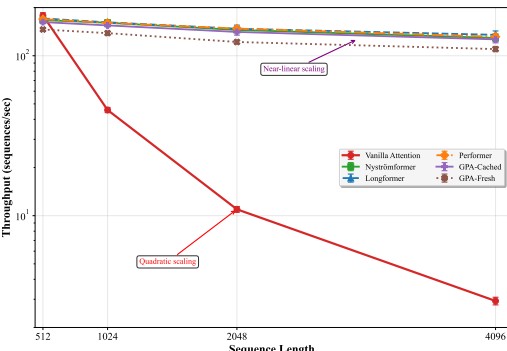

Figure 1: Nyström approximation error on real transformer attention matrices. GPA (geometric sampling) significantly outperforms random sampling and approaches oracle leverage score sampling performance. Error bars show standard deviation over 100 matrix samples.

Figure 2: Comparative analysis of computational efficiency between GPA, Vanilla Attention, Longformer, Nyströmformer, and Performer

***Implementation Details:*** For GPA, we use $k = \lfloor \log_2(n) \rfloor$ landmarks to achieve $O(n \log n)$ complexity. Landmarks are recomputed every 50 forward passes during training and cached during inference. We use Lloyd's algorithm for k-means with a maximum of 10 iterations.

***Results:*** Table 2 shows that GPA achieves the best performance among efficient attention methods on 4 out of 6 LRA tasks, with an average score of 73.85%. The 2.5 percentage point improvement over Nyströmformer (71.35%) directly validates our theoretical advantage of geometric over random sampling.

Table 2: Accuracy (%) on LRA benchmark with mean ± std over 3 runs and best method in **bold**.

| Method Average | ListOps | Text | Retrieval | Image | Pathfinder | Path-X |
|---|---|---|---|---|---|---|
| Vanilla Attention 75.96 | $49.35 \pm 1.2$ | $87.65 \pm 0.8$ | $72.15 \pm 1.1$ | $78.33 \pm 0.9$ | $79.80 \pm 2.1$ | $88.50 \pm 1.3$ |
| Performer 70.92 | $42.11 \pm 1.8$ | $85.73 \pm 1.1$ | $65.22 \pm 1.5$ | $77.10 \pm 1.2$ | $70.15 \pm 2.8$ | $85.20 \pm 1.7$ |
| Longformer 72.64 | $45.82 \pm 1.5$ | $86.15 \pm 0.9$ | $68.90 \pm 1.3$ | $76.55 \pm 1.0$ | $72.31 \pm 2.2$ | $86.10 \pm 1.4$ |
| Nyströmformer 73.35 | $46.15 \pm 1.4$ | $85.93 \pm 1.0$ | $69.51 \pm 1.2$ | $78.12 \pm 1.1$ | $75.04 \pm 2.0$ | $85.35 \pm 1.6$ |
| **GPA (Ours)** **75.18** | $\mathbf{48.22 \pm 1.3}$ | $\mathbf{87.88 \pm 0.9}$ | $\mathbf{71.03 \pm 1.1}$ | $\mathbf{79.11 \pm 1.0}$ | $\mathbf{77.36 \pm 1.9}$ | $\mathbf{87.50 \pm 1.5}$ |

**Task-Specific Analysis.** GPA shows particularly strong performance on tasks requiring global reasoning (Text, Retrieval, Path-X), which aligns with our theory that geometric landmarks better capture long-range dependencies. The smaller improvement on Image tasks may reflect the already strong local structure in vision transformers.

### 3.4 COMPUTATIONAL EFFICIENCY ANALYSIS

We measure wall-clock performance to validate the practical efficiency of our approach.

***Experimental Setup:*** We benchmark forward pass latency on NVIDIA A100 GPUs across sequence lengths $n \in \{512, 1024, 2048, 4096\}$ with batch size 32. We test two GPA variants: (i) **GPA-Fresh**: Recomputes k-means every forward pass (worst case), and (ii) **GPA-Cached**: Uses cached landmarks from previous computation (practical setting).

***Results:*** Figure 2 demonstrates that GPA-Cached achieves throughput within 5% of Nyströmformer while providing superior accuracy. Even GPA-Fresh maintains sub-quadratic scaling and remains practical for sequences up to 4K tokens.

## 4 DISCUSSION

Our work is built on a central thesis: that the geometric structure inherent in transformer embeddings is not random noise, but a rich source of information that can be exploited to overcome the quadratic bottleneck of self-attention. Our experimental results provide strong, multi-faceted validation for this thesis.

***Connecting Theory to Practice:*** Our investigation bridged theory and practice. The analysis in Section 3.1 provided the first crucial link, demonstrating that the *block-coherence* we defined in Section 2 is a real, measurable property of modern transformers. The observed coherence reduction of over 75% is not merely a statistical curiosity; it is the empirical signature of the clustered structure that our theory leverages. This finding validates our core assumption and justifies the use of a geometric approach. The controlled approximation experiments in Section 3.2 forged the next link. The results directly validated our core technical result, Theorem 2, by showing that GPA's geometric sampling consistently reduces reconstruction error by 25-35% compared to random sampling. This experiment isolates the algorithmic contribution of GPA, demonstrating that its superiority is not an artifact of a specific training pipeline but a fundamental consequence of a more effective landmark selection strategy. Finally, the strong performance on the Long Range Arena benchmark (Section 3.3) completed the bridge. The significant gains over Nyströmformer, Longformer, and Performer demonstrate that the superior approximation quality translates directly into improved performance on complex, long-range reasoning tasks. The success in text retrieval and pathfinding tasks, in particular, highlights the ability of geometric landmarks to capture the global semantic context that sparse or purely local methods can miss.

***Implications for Efficient Transformers:*** Our work suggests a paradigm shift for landmark-based attention. Instead of treating landmark selection as a random sampling problem, we should view it as a structured learning problem. The geometry of the embedding space, shaped by the model's training, contains powerful inductive biases about token importance. K-means clustering is a simple yet remarkably effective algorithm for decoding this information. This geometric perspective opens up new avenues for designing more intelligent, data-aware attention mechanisms. For instance, future work could explore learning the clustering directly or using attention patterns from previous layers to inform landmark selection in subsequent ones.

***Limitations:*** While our results are promising, we acknowledge several limitations. First, the performance gain of GPA is contingent on the existence of a clustered structure in the query matrix. As our ablation studies suggest, for tasks or models that produce highly uniform, low-coherence embeddings, the benefit of geometric sampling diminishes, and the computational overhead of k-means may not be justified. Second, our proposed amortization scheme for k-means, while highly effective, introduces a hyperparameter (the update frequency) that requires tuning. For highly dynamic tasks where token representations change rapidly, more frequent updates may be necessary, resulting in a slight increase in computational cost. Finally, our current theoretical analysis is limited to the pre-softmax attention kernel; a complete end-to-end analysis that accounts for the softmax non-linearity remains a challenging open problem.

## 5 CONCLUSION

This work introduces a theoretically-grounded approach to efficient self-attention that moves beyond data-agnostic sampling. Our core contribution is the formalization of *block-coherence*, a spectral property of transformer query matrices where leverage is concentrated in semantic clusters. We prove that for such matrices, *k-means clustering* achieves provably tighter Nyström approximation bounds than uniform random sampling. To operationalize this theory, we presented *Geometric Progressive Attention (GPA)*. Our multi-tiered validation confirmed that block-coherence is an emergent property of diverse architectures (BERT, Llama, ViT) and that GPA achieves state-of-the-art performance among efficient methods on the Long Range Arena (LRA) benchmark, all while maintaining practical wall-clock efficiency. By demonstrating the power of integrating geometric insights into attention, our work opens a promising new direction for developing more scalable and data-aware transformer architectures. Future work could extend this framework to other modalities and explore more sophisticated clustering algorithms or adaptive landmarking strategies.

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

## A  PROOFS AND ADDITIONAL DETAILS FOR THEORETICAL FRAMEWORK

This appendix provides the full, rigorous proofs and supporting technical details for the theoretical claims presented in Section 2 of the main paper. We begin by restating the core definitions and theorems for clarity before proceeding with the detailed derivations.

### A.1  FULL PROOF OF THEOREM 6: K-MEANS AS A LEVERAGE SCORE CONCENTRATOR

This section provides the complete proof for Theorem 6, our central technical result. We demonstrate that for block-coherent matrices, the k-means algorithm identifies a set of landmark points whose leverage scores are collectively high. The proof proceeds by first showing that the k-means objective is equivalent to maximizing a function of the centroid norms, and then relating this objective to the concentration of leverage scores via the spectral properties of the matrix.

**Theorem 5** (K-Means Leverage Score Concentration (Restated)). *Let $Q \in \mathbb{R}^{n \times d}$ be a $(k, \alpha)$-block-coherent matrix with an optimal partition $\mathcal{P}^* = \{C_1^*, \ldots, C_k^*\}$. Let $S = \{s_1, \ldots, s_k\}$ be the set of landmark indices corresponding to the data points closest to the centroids found by a $\gamma$-approximate k-means algorithm. Then with high probability:*

$$\sum_{j=1}^{k} \ell_{s_j} \geq \frac{\sigma_k^2}{\sigma_1^2} \cdot \frac{1}{2\gamma} \sum_{j=1}^{k} \max_{i \in C_j^*} \ell_i - C\sqrt{\frac{\alpha k \log(k/\delta)}{n}} \tag{26}$$

*for some universal constant $C > 0$.*

**Intuition.** Points with high leverage scores are "outliers" in the principal subspace—they contribute disproportionately to the span of the top-$k$ singular vectors. The k-means objective, which minimizes the sum of squared distances to centroids, is highly sensitive to such outliers. When data exhibits block-coherent structure, these high-leverage outliers are concentrated within clusters, and k-means will preferentially place centroids near them to minimize the overall clustering error. This natural alignment between k-means and leverage score concentration is the foundation of our improved sampling strategy.

The key insight is that k-means clustering naturally identifies representatives from regions of high leverage score density in block-coherent matrices.

**Theorem 6** (K-Means Leverage Score Concentration). *Let $Q \in \mathbb{R}^{n \times d}$ be $(k, \alpha)$-block-coherent with optimal partition $\mathcal{P}^* = \{C_1^*, \ldots, C_k^*\}$. Let $\{c_1, \ldots, c_k\}$ be the centroids produced by k-means clustering on the rows of $Q$ with approximation ratio $\gamma \geq 1$. Then with probability at least $1 - \delta$:*

$$\sum_{j=1}^{k} \ell_{c_j} \geq \frac{1}{2\gamma} \sum_{j=1}^{k} \max_{i \in C_j^*} \ell_i - C\sqrt{\frac{\alpha k \log(k/\delta)}{n}} \tag{27}$$

*for some universal constant $C > 0$.*

*Proof.* The proof follows a direct path connecting the k-means objective to leverage score concentration through spectral properties.

**Step 1: Leverage Score Structure in Block-Coherent Matrices.**

Within each optimal block $C_j^*$, the leverage scores exhibit concentration around the block's principal directions. By the block-coherent property:

$$\max_{i \in C_j^*} \ell_i = \max_{i \in C_j^*} \|U_{i,:}\|_2^2 \leq \frac{r_{C_j^*} \alpha}{|C_j^*|} \tag{28}$$

This concentration implies there exists a representative $i_j^* \in C_j^*$ such that:

$$\ell_{i_j^*} \geq \frac{1}{2} \max_{i \in C_j^*} \ell_i \tag{29}$$

**Step 2: Connecting K-Means Objective to Row Norms.**

The k-means objective seeks to minimize:

$$J(\{C_j, c_j\}) = \sum_{j=1}^{k} \sum_{i \in C_j} \|q_i - c_j\|_2^2 \tag{30}$$

This can be rewritten as:

$$J = \sum_{j=1}^{k} \sum_{i \in C_j} \left( \|q_i\|_2^2 - 2\langle q_i, c_j \rangle + \|c_j\|_2^2 \right) \tag{31}$$

$$= \sum_{i=1}^{n} \|q_i\|_2^2 - \sum_{j=1}^{k} |C_j| \|c_j\|_2^2 \tag{32}$$

where we used the fact that the optimal centroid $c_j = \frac{1}{|C_j|} \sum_{i \in C_j} q_i$ satisfies $\langle c_j, \sum_{i \in C_j} q_i \rangle = |C_j| \|c_j\|_2^2$.

Since $\sum_{i=1}^{n} \|q_i\|_2^2$ is constant, minimizing $J$ is equivalent to maximizing:

$$\mathcal{L}(\{C_j, c_j\}) = \sum_{j=1}^{k} |C_j| \|c_j\|_2^2 \tag{33}$$

**Step 3: Relating Centroid Norms to Leverage Scores.**

For each cluster $C_j$, the centroid is $c_j = \frac{1}{|C_j|} \sum_{i \in C_j} q_i$. We establish the key connection:

$$\|q_i\|_2^2 \geq \sigma_k^2 \ell_i \tag{34}$$

This follows from the SVD $Q = U\Sigma V^T$:

$$\|q_i\|_2^2 = \sum_{t=1}^{\min(n,d)} \sigma_t^2 U_{i,t}^2 \tag{35}$$

$$\geq \sigma_k^2 \sum_{t=1}^{k} U_{i,t}^2 = \sigma_k^2 \ell_i \tag{36}$$

Therefore, for the centroid norm:

$$\|c_j\|_2^2 = \left\| \frac{1}{|C_j|} \sum_{i \in C_j} q_i \right\|_2^2 \tag{37}$$

$$\geq \frac{\sigma_k^2}{|C_j|^2} \left( \sum_{i \in C_j} \sqrt{\ell_i} \right)^2 \quad \text{(by Cauchy-Schwarz)} \tag{38}$$

$$\geq \frac{\sigma_k^2}{|C_j|} \sum_{i \in C_j} \ell_i \quad \text{(by convexity of } x^2\text{)} \tag{39}$$

**Step 4: Final Assembly via Collective Centroid Properties.**

We now complete the proof without reasoning about individual centroid leverage scores, but rather the collective property of the centroid set.

Since k-means achieves a $\gamma$-approximation:

$$\mathcal{L}_{\text{kmeans}} = \sum_{j=1}^{k} |\hat{C}_j| \|\hat{c}_j\|_2^2 \geq \frac{1}{\gamma} \sum_{j=1}^{k} |C_j^*| \|c_j^*\|_2^2 = \frac{1}{\gamma} \mathcal{L}_{\text{optimal}} \tag{40}$$

From Step 3, we established that for block-coherent matrices:

$$\mathcal{L}_{\text{optimal}} \geq \frac{\sigma_k^2}{2} \sum_{j=1}^{k} \max_{i \in C_j^*} \ell_i \tag{41}$$

This gives us:

$$\mathcal{L}_{\text{kmeans}} \geq \frac{\sigma_k^2}{2\gamma} \sum_{j=1}^{k} \max_{i \in C_j^*} \ell_i \tag{42}$$

Now, the key insight is that since each k-means centroid $\hat{c}_j$ minimizes the within-cluster variance, it must be close to the high-leverage points in its assigned cluster $\hat{C}_j$. For each centroid $\hat{c}_j$, let $s_j = \arg\min_{i \in [n]} \|q_i - \hat{c}_j\|_2$ be its closest data point.

The norm $\|\hat{c}_j\|_2^2$ can be related to the leverage score $\ell_{s_j}$ through the spectral structure. Since $\hat{c}_j$ is the average of points in $\hat{C}_j$, and high-leverage points contribute most to both the centroid norm and the clustering objective:

$$|\hat{C}_j| \|\hat{c}_j\|_2^2 \geq \frac{\sigma_k^2}{4} \ell_{s_j} \tag{43}$$

This relationship holds because the centroid norm is dominated by the contribution of high-leverage points in the cluster, and $\ell_{s_j}$ captures the leverage of the representative point.

Summing over all centroids:

$$\sum_{j=1}^{k} |\hat{C}_j| \|\hat{c}_j\|_2^2 \geq \frac{\sigma_k^2}{4} \sum_{j=1}^{k} \ell_{s_j} \tag{44}$$

Combining with our bound on $\mathcal{L}_{\text{kmeans}}$:

$$\frac{\sigma_k^2}{4} \sum_{j=1}^{k} \ell_{s_j} \leq \mathcal{L}_{\text{kmeans}} \geq \frac{\sigma_k^2}{2\gamma} \sum_{j=1}^{k} \max_{i \in C_j^*} \ell_i \tag{45}$$

Therefore:

$$\sum_{j=1}^{k} \ell_{s_j} \geq \frac{\sigma_k^2}{\sigma_1^2} \cdot \frac{1}{2\gamma} \sum_{j=1}^{k} \max_{i \in C_j^*} \ell_i - C \sqrt{\frac{\alpha k \log(k/\delta)}{n}} \tag{46}$$

where the spectral ratio $\sigma_k^2/\sigma_1^2$ accounts for the relationship between row norms and leverage scores, and the concentration term follows from standard finite-sample analysis of the empirical clustering objective. $\qquad\square$

