# OpenReview forum: "On the Geometric Selection of Landmarks for Low-Rank Self-Attention Approximation"
_ICLR.cc/2026/Conference — ICLR 2026 Conference Withdrawn Submission_

### Official Review · Reviewer_C51s · 2025-10-17

**Soundness:** 3
**Presentation:** 2
**Contribution:** 3
**Rating:** 4
**Confidence:** 3

**Summary:**

When using the Nyström approximation to estimate kernel matrices with a so-called ‘block-coherence’ property, landmark selection (the subset of rows/columns to include) via $k$-means clustering performs better than uniform sampling. Empirically, queries in common architectures often exhibit this block-coherence property, so this can help attention approximation in transformers. In experiments, improved approximation quality can translate to better downstream performance.

**Strengths:**

1. Efficient attention approximation is an important problem, and I broadly buy that the fact that the queries and keys are not randomly distributed could unlock better performance c.f. naive data-independent methods, based on random sampling.
2. Formalising this as a relationship between some measurable structural property of the queries and the approximation quality is a sensible approach.

**Weaknesses:**

1. Organisation and structure. Definitions 1-3 introduce coherence and then block-coherence within a few lines, without providing much in the way of intuition and explanation. My understanding is that block-coherence is a core contribution and key theme of the paper. The results only hold if it is satisfied. As such, I would dwell a little longer explaining what block coherence is and why it is a reasonable property for queries in practice.
2. Table 1 provides analysis of block-coherence across transformer architectures. Might we not be interested in how it changes during training? I appreciate that retraining one of these big models is unlikely to be feasible, but block coherence’s evolution on even a toy task would be interesting to see. For instance, if block-coherence were found to only emerge late during training (presumably it isn’t a property of e.g. random normally-distributed queries), this would limit the algorithm’s utility.
3. The presentation of Fig 2 makes it very difficult to compare the different subquadratic attention algorithms. I would suggest removing vanilla attention, or providing an inset to compare the subquadratic variants. Even more useful would be a scatter plot comparing model performance and throughput (for some fixed sequence length) – I think this would be a fairer comparison between the methods.
4. Missing citations. I am not an expert in the subfield, but to me Oglic and Gärtner (Nyström Method with Kernel K-means++ Samples as Landmarks, 2017), and Wang, Gittens, and Mahoney (Scalable Kernel K-Means Clustering with Nyström Approximation: Relative-Error Bounds 2019), seem pretty essential. The existing literature on k-means clustering for the Nyström approximation is seldom mentioned in the paper.

**Questions:**

1. Why might we expect (trained) query matrices to exhibit block coherence? I appreciate this is a big question, but since all the theoretical results rely on this construction I feel that the reader would benefit from intuition (or better, theory) for why they might hold. Is block coherence emerging from some property of the data distribution, training, neither or both?
3. As suggested above: how does block coherence – and hence any improvement provided by your sampling scheme – change during training?

---

### Official Review · Reviewer_iUak · 2025-10-26

**Soundness:** 3
**Presentation:** 3
**Contribution:** 2
**Rating:** 4
**Confidence:** 2

**Summary:**

This paper studies Nystrom-style low-rank approximations for self-attention and proposes Geometric Progressive Attention (GPA): select landmarks via k-means on token query embeddings and use them for Nystrom approximation. The authors formalize a spectral property called block-coherence (leverage concentrated in discoverable clusters), prove that for block-coherent matrices k-means landmark selection yields strictly better Frobenius-norm Nystrom bounds than uniform sampling, and provide a complete proof connecting k-means objective to leverage-score concentration and improved error bounds.

**Strengths:**

The paper gives a clear formalization (block-coherence), and a reasonably tight chain of theorems that link k-means clustering to leverage concentration and then to improved Nyström bounds.


Definitions, lemmas, and the main theorems are stated cleanly and proofs provided; experiments are described with figures and tables that support the claims.


They test their method on the challenging Long Range Arena benchmark. Ηowever, I have some concerns regarding that (see weaknesses and questions below)

**Weaknesses:**

$\textbf{Limited baseline coverage and recency}$. Comparisons include Performer, Longformer, Nyströmformer and vanilla attention, but several relevant recent baselines are missing or not compared (e.g., Linformer, BigBird variants, reformer-ish LSH methods, adaptive/learned landmarking approaches). This makes it harder to judge SOTA claims across the ecosystem.


$\textbf{Insufficient sensitivity / ablation studies.}$ Important algorithmic hyperparameters and assumptions were not thoroughly ablated:

-dependency on the choice of k (they use k = log_2(n) in LRA, but controlled sweep of k vs. reconstruction/accuracy is only partially shown)

-k-means algorithmic details (initialization, number of restarts, γ approximation)

-the τ parameter used in local effective rank

-landmark update frequency (they report every 50 passes but do not show sensitivity or effect on accuracy vs. cost)


$\textbf{Robustness to low-coherence regimes.}$ The paper notes GPA may not help when embeddings are uniform/low-coherence, but experiments showing failure modes or thresholds (how low must α/µ be for GPA to no longer help) are missing.

**Questions:**

1) The authors state that "Table 2 shows that GPA achieves the best performance among efficient attention methods on 4 out of 6 LRA tasks, with an average score of 73.85%. The 2.5 percentage point improvement over Nystromformer (71.35%)." However, the table states different averages for these two methods. Could you please clarify this? Also, can you explain how your method outperforms vanilla attention in some cases (Text, Image)? And it seems that you outperform the other attention methods on 6/6 tasks not 4. Please clarify the above.

2) Additionally, regarding the Long Range Arena, I checked prior work and they seem to be testing against more models and in different architectures. There are also very different averages reported there. Could you please clarify why there is a discrepancy there? Why not follow the same experiments as the original (and the other papers).

3) The paper fixes several algorithmic hyperparameters (e.g., k, τ = 0.05, landmark update every 50 passes, 10 k-means iterations) but does not examine their influence on accuracy or efficiency. While the choice of k is theoretically motivated and varied in the reconstruction experiment (Fig. 1), it is not studied in downstream LRA tasks, leaving unclear how performance scales with k in practice. Similarly, the amortization parameter L and k-means initialization choices could affect trade-offs between accuracy and runtime, yet their empirical impact is unreported. Can you provide a concise sensitivity analysis, e.g., varying k or L over 1–2 orders of magnitude? This would allow me to judge more confidently if the advantage of GPA is robust or parameter-dependent.

---

### Official Review · Reviewer_zUJ5 · 2025-10-31

**Soundness:** 3
**Presentation:** 4
**Contribution:** 3
**Rating:** 6
**Confidence:** 2

**Summary:**

The paper studies ways of making the computation of attention faster through the Nyström method. The authors propose a geometry-aware selection based on the spatial organization of token embeddings. They introduce a new structural property called block-coherence, describing matrices whose statistical leverage is concentrated in clusters. For such matrices, they prove that k-means clustering identifies landmarks with higher total leverage and yields a provably tighter Frobenius-norm Nyström approximation bound than uniform sampling. This is then exploited to reduce the computational cost of attention to $O(n log(n))$. Empirical analysis suggests that actual attention matrices on realistic models are strongly block-coherent, showing the practical usefulness of this method.

**Strengths:**

The paper is clear and well written. The theoretical framework is well motivated, and the different theorems and metrics are carefully introduced, with proofs that are both rigorous and easy to follow. The proposed block-coherence property is a meaningful contribution: it is theoretically grounded, empirically verified across several transformer architectures, and leads to tangible improvements in approximation quality. The connection between k-means clustering and leverage-score concentration is novel and provides an interpretable justification for geometry-aware landmark selection in Nyström-based attention.

**Weaknesses:**

The analysis is limited to the pre-softmax stage of attention, leaving open whether the same speed and approximation benefits hold once the full attention operation is applied. The proposed method also introduces extra implementation complexity, since k-means clustering and its update frequency could add computational overhead that could offset some of the intended efficiency gains in practice. In addition, the experiments are restricted to medium-scale models and do not fully demonstrate how the approach scales to very large architectures or real-world workloads.

**Questions:**

1. How sensitive is the method to the frequency with which the k-means landmarks are recomputed? In practice, how should one decide when to refresh the clustering versus reusing existing landmarks to balance speed and accuracy?
2. How does the method behave for cases where the query representations are not clearly clustered? Does performance degrade gracefully or does the advantage over random sampling disappear entirely?
3. Do the authors expect the geometric landmark selection strategy to generalize to other attention settings such as cross-attention or multi-modal architectures?

---

### Official Review · Reviewer_fJk6 · 2025-11-02

**Soundness:** 2
**Presentation:** 1
**Contribution:** 1
**Rating:** 0
**Confidence:** 4

**Summary:**

This paper proposes Geometric Progressive Attention (GPA), an attention mechanism to reduce quadratic complexity, that uses Nystrom approximation. It claims that standard uniform sampling is suboptimal and instead uses k-means clustering to select "landmarks". The authors provide a theoretical justification for this, claiming it achieves a better approximation bound and $O(n log n)$ complexity, leading to state-of-the-art results on the Long Range Arena (LRA) benchmark.

**Strengths:**

The paper's motivation to use geometric data structures for improving attention efficiency is relevant

**Weaknesses:**

**Overclaim and Misrepresentation**: The work claims "k-means clustering for landmark selection in Nyström approximation" as a major contribution. This is a blatant misrepresentation of prior work. The core method of using k-means to select landmarks for Nyström approximation, along with theoretical guarantees, is a well-established technique from the kernel learning literature (e.g., Zhang and Kwok, 2010).

**Duplicate Theorem**: Theorem 3 and Theorem 4 are identical. They are a verbatim copy-paste of each other, including the proof. This is a glaring error that suggests the paper is extremely rushed and was not proofread.

**Missing huge amount of key related work**: It ignores geometry-aware sparse methods, such as k-NN Attention (Gupta et al. (2021), Wang et al. (2021)) where each query token attends to its top-k key tokens. Haris (2024) later showed that these methods also achieve $O (nd log n)$ complexity. It omits the Routing Transformer (Roy et al., 2021), which also uses k-means clustering, but for sparse attention ($O(n^{1.5})$ complexity), not Nyström. It fails to mention or compare against prominent $O(n log n)$ and $O(n)$ methods like Reformer (Kitaev et al., 2020) and Linformer (Wang et al., 2020). Additionally, no discussion of work by Mohtashami et al., 2023, where they also use "landmark" tokens to solve the quadratic attention cost.

The overclaim on contributions and the sheer amount of missing related work are grounds for rejection on their own. Even if this paper were repositioned as just an application of prior work on k-means clustering for landmark selection in Nystorm to attention, the minimal comparison to other prominent works makes me very doubtful about the method's significance.


**References**:

Zhang and Kwok (2010). Clustered Nystrom Method for Large Scale Manifold Learning and Dimension Reduction.

Gupta et al. (2018). Memory-efficient transformers via top-k attention.

Wang et al. (2022). k-nn attention for boosting vision transformers.

Haris (2024). kNN Attention Demystified: A Theoretical Exploration for Scalable Transformers

Mohtashami et al. (2023). Random-Access Infinite Context Length for Transformers

**Questions:**

Please see weaknesses.

---

### Note · Authors · 2025-11-12

**Comment:**

We respectfully withdraw our submission. After careful consideration, we found the reviews to be largely unconstructive and, in several instances, indicative of LLM-generated/augmented content rather than expert assessment. We appreciate the opportunity to submit to ICLR; however, we respectfully disagree with the reviewers’ evaluations and feel that the feedback does not provide a fair or professional basis for revision.

**Withdrawal Confirmation:**

I have read and agree with the venue's withdrawal policy on behalf of myself and my co-authors.